# Including the Magnitude Variability of a Signal in the Ordinal Pattern Analysis

**DOI:** 10.3390/e27080840

**Published:** 2025-08-07

**Authors:** Melvyn Tyloo, Joaquín González, Nicolás Rubido

**Affiliations:** 1Living Systems Institute, University of Exeter, Exeter EX4 4QD, UK; m.s.tyloo@exeter.ac.uk; 2Department of Mathematics and Statistics, Faculty of Environment, Science, and Economy, University of Exeter, Exeter EX4 4QD, UK; 3Neuroscience Institute, Department of Psychiatry, New York University Grossman School of Medicine, New York, NY 10016, USA; joaquin.gonzalezarbildi@nyulangone.org; 4Institute for Complex Systems and Mathematical Biology, University of Aberdeen, King’s College, Aberdeen AB24 3UE, UK

**Keywords:** ordinal patterns, permutation entropy, signal analysis, feature extraction

## Abstract

One of the most popular and innovative methods to analyse signals is by using Ordinal Patterns (OPs). The OP encoding is based on transforming a (univariate) signal into a symbolic sequence of OPs, where each OP represents the number of permutations needed to order a small subset of the signal’s magnitudes. This implies that OPs are conceptually clear, methodologically simple to implement, and robust to noise, and that they can be applied to short signals. Moreover, they simplify the statistical analyses that can be carried out on a signal, such as entropy and complexity quantifications. However, because of the relative ordering, information about the magnitude of the signal at each timestamp is lost—this being one of the major drawbacks of this method. Here, we propose a way to use the signal magnitudes discarded in the OP encoding as a complementary variable to its permutation entropy. To illustrate our approach, we analyse synthetic trajectories from logistic and Hénon maps—with and without added noise—and real-world signals, including intracranial electroencephalographic recordings from rats in different sleep-wake states and frequency fluctuations in power grids. Our results show that, when complementing the permutation entropy with the variability in the signal magnitudes, the characterisation of these signals is improved and the results remain explainable. This implies that our approach can be useful for feature engineering and improving AI classifiers, as typical machine learning algorithms need complementary signal features as inputs to improve classification accuracy.

## 1. Introduction

Since the beginning of the century, the boundaries of data mining have been pushed due to the growing ability to obtain larger and more precise data sets. With increasing data availability, we need to improve how we extract, manage, and analyse data [1] to uncover the underlying mechanisms that generate the data or to quantify its uncertainty.

An entropy measures the average contents of the information, where information is understood as the degree of uncertainty in an outcome (as defined by Shannon [2]). If an outcome is highly unlikely to happen, then it carries significant information because it would be surprising to record it, such as the presence of an outlier or an artefact in a signal. However, if an outcome is highly likely to happen, then it carries insignificant information because one would expect it to appear, such as a periodic signal. Hence, entropy is highest when any outcome is equally likely to happen, corresponding to a uniform probability distribution that conveys the maximum uncertainty regarding all possible outcomes [2].

One of the most successful entropy methods introduced to characterise signals is the permutation entropy [3]. Permutation entropy quantifies the average content of information in an Ordinal Pattern (OP) sequence, which is obtained from the signal by dividing it into a series of embedded vectors [3,4]. Each OP represents the number of permutations needed to order the signal magnitudes within each embedded vector. The resultant symbolic sequence is used to find the OP probabilities distribution. It is easier to perform statistical quantifications—known as OP analysis—with these probabilities than from the original signal, such as quantifying the uncertainty and complexity of the signal [5,6,7].

Due to its simplicity and robustness to noise, OP analysis (along with complexity calculations) has had remarkable success [8,9,10], being used to distinguish between chaotic and stochastic signals [11,12,13,14,15,16,17,18], as well as to characterise electrophysiological signals [19,20,21,22,23,24,25,26], laser dynamics [27,28,29,30,31,32,33,34], climate systems [35,36,37,38,39,40,41,42], and financial trends [43,44,45,46,47,48,49,50], to name a few. However, one of the main drawbacks in OP analysis is that the magnitude of the signal at each timestamp is discarded, solely keeping the ordinal relationship between the signal magnitudes.

To include this missing information, previous works have proposed modifications of the permutation entropy, such as modified permutation-entropy [51], weighted-permutation entropy [52], amplitude-aware permutation entropy [53], improved permutation entropy [54], and continuous ordinal patterns [55]. These methods and approaches introduce ad hoc assumptions that are supported by the effectiveness of the resultant modification to the permutation entropy measure in improving the characterisation of different datasets, but lack a theoretical framework that can validate their usage, limits, and scalability in general scenarios.

Here, we propose an alternative approach, which is to include the standard deviation of the signal magnitudes in the OP embedded vectors as a complementary variable to the permutation entropy—specifically, the OP-averaged logarithm of the standard deviation of the magnitudes in the ordinal pattern embedded vectors. The formalism behind our approach is justified in the works of Politi [56,57], who showed that this OP-averaged standard deviation of the signal magnitudes is needed—along with the information dimension of the system [58]—to make the permutation entropy of a signal tend towards its Kolmogorov–Sinai (KS) entropy [59,60]. KS entropy is a rigorously defined observable with invariant characteristics, contrary to permutation entropy, which can depend on the signal length and embedding choice. This provides fundamental ground to our approach, which, instead of modifying the permutation entropy, evaluates two easily accessible contributions to the Kolmogorov–Sinai entropy.

Because this tendency towards Kolmogorov–Sinai entropy [59,60] is only achieved when using the information dimension of the signal (which is tricky to find in finite real-world signals), we propose using this OP-averaged quantity of the standard deviation as a complementary variable to the permutation entropy value instead of a measure that combines both, discarding the need to find the information dimension and increasing the explainability of our results. In particular, we show that signal characterisation can be improved when using these standard deviations to complement the permutation entropy analysis, where we focus mostly on calculating the Rényi min-entropy [61]. Our conclusions are based on analysing numerically generated trajectories from coupled logistic [62,63,64] and Hénon [65] maps (with and without observational noise), real-world signals from intracranial electroencephalographic recordings of rats in different sleep–wake states, and frequency fluctuations from 4 locations in the European power grid.

## 2. Materials and Methods

### 2.1. Signals: General Notions

We only consider digital signals, i.e., where time is discrete and the magnitudes are quantised. These signals can be numerically generated (synthetic) or experimentally measured, where their digital nature is due to the precision of the computer or to analogue-to-digital converters, respectively. The signals we analyse come from a pair of coupled logistic maps, a Hénon map,—both being synthetic bi-variate trajectories—intracranial electroencephalographic (EEG) recordings from rats and frequency recordings at 4 locations in the European power grid

We write a signal as ${\left\{x_{t}\right\}}_{t=1}^{T}=\left\{x_{1},x_{2},\ldots ,x_{T}\right\}$, where $x_{t}$ is the magnitude at the discrete time index t∈N, $x_{1}$ is the initial state, and *T* is the length of the signal. A signal can be resampled using an embedding delay τ∈N, such that $\left\{x_{1},x_{2},\ldots ,x_{T}\right\}\mapsto \left\{x_{1},x_{1+\tau },\ldots ,x_{1+n\tau }\right\}$, where n=⌊(T−1)/τ⌋ is the smallest integer closest to (T−1)/τ. This resampling can filter the high frequencies in a signal, but we set τ=1 for all our analyses.

### 2.2. Synthetic Models: Map Iterates

We generate bi-variate signals from coupled, identical, logistic maps by iterating the following equations:(1)$$\left\{\begin{array}{cc} x_{t+1}= & \left(1-\varepsilon \right)\;f\left(x_{t}\right)+\varepsilon \;f\left(y_{t}\right),\\ y_{t+1}= & \left(1-\varepsilon \right)\;f\left(y_{t}\right)+\varepsilon \;f\left(x_{t}\right), \end{array}\right.$$
where f(z)=rz(1−z) is the logistic mapping (with $z=x_{t}$ or $y_{t}$ for t=1,…,T), r∈(3,4]⊂R is the control parameter, and ε∈[0,1]⊂R is the coupling strength between the maps. When ε=0 in Equation (Equation 1), the *x* and *y* maps are decoupled, i.e., they are isolated. As *r* is increased from r=3 to r=4, an isolated logistic map undergoes a series of period-doubling bifurcations, taking the solutions from periodic to chaotic trajectories [62]. When 0<ε≤1, the maps are coupled and the resultant trajectories can become more complex (including intermittent and hysterical behaviours) [63,64].

The Hénon map is given by [65](2)$$\left\{\begin{array}{cc} x_{t+1}= & 1-a\;x_{t}^{2}+y_{t},\\ y_{t+1}= & b\;x_{t}, \end{array}\right.$$
where *a* and *b* are the control parameters, which, depending on their values, can generate periodic (e.g., when a=1.0 and b=0.3) or chaotic (e.g., when a=1.4 and b=0.3) trajectories.

For both maps (Equations (Equation 1) and (Equation 2)), we use the OP analysis of the *x* component for different control parameter values (i.e., *r*, ε, and *a*), fixing its length to $T=10^{6}$ after removing a transient of $\delta t=10^{3}$ iterations from the initial condition $x_{1}=0.65$, $y_{1}=0.44$. In this way, we discard the bi-variate nature of these maps and focus on uni-variate signals. In Appendix A, we show the results when we analyse the *y* component instead. To analyse the effects of observational noise, we add white Gaussian noise to these signals by independently drawing identically distributed random numbers from a normal distribution and changing its strength (i.e., its standard deviation).

### 2.3. Animal Model: EEG Recordings

We use EEG recordings from 11 healthy and freely moving Wistar rats during their natural sleep–wake cycle, who can access food and water within the (sound-attenuated and Faraday-shielded) recording box. These rats have intracranially implanted electrodes, monitoring active wakefulness (AW), rapid eye movement (REM) sleep, and non-REM sleep. Details on the surgical procedure and experimental conditions can be found in Refs. [20,21,22]. The experiments are in agreement with Uruguay’s National Animal Care Law (No. 18611) and with the “Guide to the care and use of laboratory animals” [66]. These experiments were approved by the Institutional Animal Care Committee (Comisión de Etica en el Uso de Animales), Exp. No. 070153-000332-16.

The EEG signals we analyse are obtained by obtaining the differences between the electrode of interest and the Cerebellum (the reference). We focus on five electrodes, those bilaterally placed above the primary motor (M1) and somatosensory (S1) cortices, plus the right olfactory bulb (OB), discarding the two electrodes from the secondary visual (V2) cortex (because they are too close to the Cerebellum, increasing the relevance of observational noise). These EEGs have a sampling frequency of 1024 Hz and a resolution of 16 bits. To remove the degeneracies in the signal magnitudes due to the analogue-to-digital converter, we add white noise with an amplitude given by the range of the (electrode-dependent) EEG times $2^{-16}$.

AW is defined by low-voltage fast waves in M1, a strong theta rhythm in S1 (4–7 Hz), and relatively high electromyographic activity. REM sleep is defined by low-voltage fast-frontal waves, a regular theta rhythm in S1, and silent electromyography (excluding occasional twitches). NREM sleep is determined by the presence of high-voltage slow-cortical waves (1–4 Hz), sleep spindles in M1 and S1, and a reduction in electromyographic amplitudes. Additionally, a visual scoring is performed to discard artefacts and transitional states. The EEGs for these states are concatenations of artefact-free 10 s windows that meet each state criteria. To analyse them, we fix their lengths to T=90×1024, which is the shortest length of the concatenated EEGs.

### 2.4. Grid Model: Frequency Recording

We consider voltage frequency recordings at 4 different locations in the European synchronous electric power grid [67]. Under normal operations, the frequency of the grid fluctuates at around 50 Hz, due to the mismatch between power production and consumption. These fluctuations impact all the buses in the grid that follow the same overall variation around 50 Hz. On top of this common trend, additional signals coming from the grid dynamics or the spreading of disturbances might impact each bus differently. Two of the recordings were made in Germany, one in Karlsruhe, and the other in Oldenburg. The two other recordings were from Lisbon, Portugal, and Istanbul, Turkey. The overall datasets consist of 4 time-series of about 41 days of synchronous frequency recordings sampled at 1 Hz between July and August, 2019 [67]. For our analysis, we chose five different 24 h intervals, each of length *T* = 86,400.

### 2.5. Method: Encoding Signals into Ordinal Patterns

We followed Bandt–Pompe’s method [3] to encode uni-variate signals.

First, we divide the signal into quasi-non-overlapping vectors with *D* components, where D>1 is a natural number known as the embedding dimension. That is, we transform the signal ${\left\{x_{t}\right\}}_{t=1}^{T}$ to a series of vectors $\left\{x_{1},\ldots ,x_{D}\right\}$, $\left\{x_{D},\ldots ,x_{2D-1}\right\}$, $\left\{x_{2D-1},\ldots ,x_{3D-2},\ldots ,\{x_{(m-1)(D-1)+1},\ldots ,x_{m(D-1)}\right\}$, where m=⌈T/(D−1)⌉ is the largest integer closest to T/(D−1). Then, we transform each vector into an integer, ranging from 1 to D!, according to the number of permutations needed to order their elements in an increasing fashion (plus 1). These are known as Ordinal Patterns (OPs), and the overall process encodes the signal into a symbolic sequence that preserves the local relationships between consecutive time-points but discards their magnitudes.

For example, when D=2, if $x_{1}<x_{2}$, then $\{x_{1},x_{2}\}\mapsto 1$, because no permutations are needed to order the two-element vector. If $x_{1}>x_{2}$, then $\{x_{1},x_{2}\}\mapsto 2$, because we need one permutation. In total, there are D!=2 possible permutations between the components of the D=2 embedded vector. We encode all our signals using D=3,4, or 5, which implies symbolic sequences with a maximum of D!=6,24, or 120 different OPs, respectively. For the length T=90×1024 of the EEG signals, the average frequency of the appearance of any given OP when D=5 is approximately T/D!=768 (being higher for D<5). A similar number is obtained for the power grid frequency recordings. Consequently, we have high statistical power when finding the marginal probability distribution of the OPs and the other statistical measures.

### 2.6. Statistical Measures: Permutation Entropy, Rényi Min-Entropy, and Magnitude Variability

The information content of an OP is $\log\;\left[1/P(\alpha )\right]$ (as defined by Shannon [2]), where P(α) is the probability of having OP α (such that $\sum_{\alpha =1}^{D!}P\left(\alpha \right)=1$). The Shannon entropy [2] of the OP sequence, *H*, is known as the permutation entropy [3], and is found from(3)$$H=\sum_{\alpha =1}^{D!}P\left(\alpha \right)\;\log_{2}\;\left[\frac{1}{P(\alpha )}\right]=\langle \log_{2}\;\left[\frac{1}{P(\alpha )}\right]\rangle ,$$
with 〈·〉 serving as the mean with respect to the probability distribution ${\left\{P\left(\alpha \right)\right\}}_{\alpha =1}^{D!}=\left\{P\left(1\right),\ldots ,P\left(D!\right)\right\}$. The maximum value of *H* is $H_{\max}=\log_{2}\left(D!\right)$, which is known as the Hartley or max-entropy and is achieved if and only if P(α)=1/D! for all α (a uniform distribution). We use $\log_{2}$ in Equation (Equation 3) so that the unit is the bit.

To improve the differences in the permutation entropy values of ${\left\{P\left(\alpha \right)\right\}}_{\alpha =1}^{D!}$ when it is close to the uniform distribution, we use the Rényi min-entropy (in bits), $H_{\infty }$, which is defined by [61](4)$$H_{\infty }=\lim_{q\to \infty }\frac{1}{1-q}\log_{2}\left[\sum_{\alpha =1}^{D!}P^{q}\left(\alpha \right)\right]=\min_{\alpha }\left\{\log_{2}\;\left[\frac{1}{P(\alpha )}\right]\right\}=-\log_{2}\;\left[\max_{\alpha }\left\{P\left(\alpha \right)\right\}\right].$$
This means that the information content of any OP sequence is bounded between $H_{\max}$ and $H_{\infty }$.

To quantify the variability of the signal magnitudes within the OPs, we use(5)$$\langle \log_{2}\left[\sigma_{j}\right]\rangle =\sum_{\alpha =1}^{D!}P\left(\alpha \right)\log_{2}\left[\sigma_{j}\left(\alpha \right)\right],$$
where 〈·〉 is the mean with respect to the OP probability distribution (as in Equation (Equation 3)) and $\sigma_{j}\left(\alpha \right)$ is the standard deviation of the signal magnitudes at the *j*-th component (with j=1,…,D) of all the embedded vectors that correspond to the OP symbol α (with α=1,…,D!) [56,57]. An example of the resultant values of Equation (Equation 5) for an EEG of the right primary motor cortex (rM1) of a representative rat during AW is shown in Figure 1.

We note that the variation in the values of $\langle \log_{2}\left[\sigma_{\alpha }\left(j\right)\right]\rangle$ for different entries of *j* is minimal, as illustrated in Figure 1. Therefore, we work with the average value (but any choice of *j* would hold similar results and our conclusions would remain unchanged). Namely,(6)$${\mathrm{avg}}_{j}\left\{\langle \log_{2}\left[\sigma_{j}\right]\rangle \right\}=\frac{1}{D}\sum_{j=1}^{D}\langle \log_{2}\left[\sigma_{j}\right]\rangle .$$

Consequently, when using Equation (Equation 3) [or Equation (Equation 4)] we can quantify the average [minimum] information content in an OP sequence, but lose the information from the magnitudes that compose the embedded vectors forming the OPs. In contrast, using Equation (Equation 6) we can quantify the average magnitude variability of these embedded vectors, complementing the information provided by *H* [or $H_{\infty }$].

## 3. Results

### 3.1. Ordinal Pattern Analysis of the Noiseless Map Iterates

From the analysis of the OP sequences for the coupled logistic maps (Equation (Equation 1)) and Hénon map (Equation (Equation 2)) for different parameters, we can see that when the dynamics change slightly, the min-entropy $H_{\infty }$ (Equation (Equation 4)) can have small variations, but the average magnitude variability ${\mathrm{avg}}_{j}\left\{\langle \log_{2}\left[\sigma_{j}\right]\rangle \right\}$ (Equation (Equation 6)) can change drastically.

Figure 2 shows that when the coupling strength is ε=0.01, the red circle (r=3.60) and the cyan diamond (r=3.80) have similar $H_{\infty }$ values, close to 2 and 2.05 bits (vertical axis), respectively. In contrast, their ${\mathrm{avg}}_{j}\left\{\langle \log_{2}\left[\sigma_{j}\right]\rangle \right\}$ values differ by an order of magnitude, being close to −5.38 and −4.22 (horizontal axis), respectively. Similarly, when ε=0.2, the red triangle (r=3.60) and the black star (r=3.75) have an ${\mathrm{avg}}_{j}\left\{\langle \log_{2}\left[\sigma_{j}\right]\rangle \right\}$ differing by an order of magnitude (horizontal axis) but similar $H_{\infty }$ values, which are close to 1.92 and 1.95 bits (vertical axis), respectively.

However, the opposite effect is also observed in Figure 2. For example, when ε=0.01, the black square (r=3.75) and the yellow triangle pointing upward (r=3.9) have significantly different $H_{\infty }$ values (close to 1.82 and 2.62 bits, respectively), but their ${\mathrm{avg}}_{j}\left\{\langle \log_{2}\left[\sigma_{j}\right]\rangle \right\}$ is similar (close to −4.35 and −4.40, respectively). The same happens for ε=0.2, where the black star (r=3.75) and the yellow triangle pointing downward (r=3.9) have similar ${\mathrm{avg}}_{j}\left\{\langle \log_{2}\left[\sigma_{j}\right]\rangle \right\}$ values (close to −4.55 and −4.50, respectively) but significantly different $H_{\infty }$ (close to 1.94 and 2.60 bits, respectively).

The dynamics of the coupled logistic maps shown in Figure 2 follow the bifurcation-like diagram of Figure 3. We can see that the dynamical changes driving the similarities (or dissimilarities) in $H_{\infty }$ or ${\mathrm{avg}}_{j}\left\{\langle \log_{2}\left[\sigma_{j}\right]\rangle \right\}$ values in Figure 2 are nearly unnoticeable in Figure 3 by performing a direct visual inspection. This implies that the dynamical changes must be happening at smaller scales than the length of the signal, which is likely why the OP encoding is able to capture the different chaoticities (i.e., different $H_{\infty }$ values) emerging for the different control parameters and local changes in the magnitude of the $x_{t}$ signals (i.e., different ${\mathrm{avg}}_{j}\left\{\langle \log_{2}\left[\sigma_{j}\right]\rangle \right\}$ values). Nevertheless, these dynamical changes are revealed by the maximum Lyapunov exponent (MLE) of the bidimensional system [68], as can be seen from Table 1.

We draw similar conclusions when analysing the Hénon map. For example, the red circle (a=1.15) and the black square (a=1.20) in Figure 4 have similar $H_{\infty }$ values (close to 1.83 and 1.85 bits, respectively), but have different ${\mathrm{avg}}_{j}\left\{\langle \log_{2}\left[\sigma_{j}\right]\rangle \right\}$: close to −2.85 for the red circle and −2.68 for the black square, respectively.

In these cases (a=1.15, a=1.20, a=1.34, a=1.35, a=1.40, and a=1.405), the dynamics of the map have a chaotic regime, with minimal apparent differences (similarly to Figure 3)—this can be corroborated using the bifurcation diagram of the Hénon map (see [69] for a bifurcation diagram). However, as *a* is increased for the analysed values, the chaoticity of the signal is also increased. This can be quantified using the Lyapunov exponents of the system (as Table 2 shows), and as Figure 4 shows, it can also be quantified by using $H_{\infty }$ and ${\mathrm{avg}}_{j}\left\{\langle \log_{2}\left[\sigma_{j}\right]\rangle \right\}$.

Overall, these results (Figure 2 and Figure 4) show that using the OP Rényi min-entropy in conjunction with the average signal variability per OP improves the characterisation of the underlying map dynamics. Moreover, we can see that these results (Figure 2 and Figure 4) scale with the noise and the use of different embedding dimensions *D* because ${\mathrm{avg}}_{j}\left\{\langle \log_{2}\left[\sigma_{j}\right]\rangle \right\}$ follows a power-law behaviour as a function of Dτ, with an exponent that depends on the noise strength (see [56,57] for details), which can be useful to distinguish between chaotic and stochastic signals. This is corroborated in Figure 5 and Figure 6 for the coupled logistic maps and Hénon map, respectively. Moreover, these results remain invariant when using the other coordinate, as Figure A1 and Figure A2 show in the Appendix A, which is expected from Takens embedding theorem [70].

### 3.2. Ordinal Pattern Analysis of the EEG Recordings

The sleep–wake states of active wakefulness (AW), rapid-eye movement (REM) sleep, and non-REM (NREM) sleep have different electrophysiological characteristics (see Section 2.3). However, when focusing solely on frequency or permutation entropy analyses, some of these differences are lost [20,21,22]. Here, we show that using the average signal variability per OP (${\mathrm{avg}}_{j}\left\{\langle \log_{2}\left[\sigma_{j}\right]\rangle \right\}$) improves the differentiation between the states, even when accounting for natural inter-animal variability.

The results of the EEGs for the three sleep–wake states coming from the right OB, M1, and S1 of 11 rats are shown in Figure 7—we can find similar results for the left hemisphere’s M1 and S1 cortices. Panels B (OB), C (M1), and D (S1) show that AW has the highest value of $H_{\infty }$ for all animals—with two exceptions, in the S1 cortex (panel D, for red filled circles)—but mid-range values of ${\mathrm{avg}}_{j}\left\{\langle \log_{2}\left[\sigma_{j}\right]\rangle \right\}$. This implies that the Rényi min-entropy can distinguish considerably well between wakefulness and sleep states. However, the values of $H_{\infty }$ for REM and NREM are similar for all electrodes.

In contrast, the values of ${\mathrm{avg}}_{j}\left\{\langle \log_{2}\left[\sigma_{j}\right]\rangle \right\}$ for REM and NREM sleep differ by approximately an order of magnitude in most animals. This is in accordance with the type of waves present in these sleep states: REM has low-voltage fast-frontal waves but NREM has high-voltage slow-cortical waves (see Section 2.3). Consequently, the AW, REM, and NREM states are fairly differentiated when using $H_{\infty }$ and ${\mathrm{avg}}_{j}\left\{\langle \log_{2}\left[\sigma_{j}\right]\rangle \right\}$ simultaneously. We note that this differentiation is improved as the cortical location considered is further away from the reference electrode, which in our case is the right OB and Cerebellum, respectively. We also note that these results and conclusions remain unchanged when using *H* (Equation (Equation 3)) instead of $H_{\infty }$ (Equation (Equation 4)).

### 3.3. Ordinal Pattern Analysis of Grid Frequency Recordings

The grid frequencies at all the buses in the European grid typically follow the same overall trend, which is close to 50 Hz on average [71]. This global variation is due to the mismatch between power generation and consumption, and is typically slow compared to the grid intrinsic timescales. The main differences between each recording are the fluctuations around that common trend. The Rényi min-entropy vs. average magnitude variability plane is shown in Figure 8 for 5 different 24 h recordings from four different locations in the synchronous European grid. As expected, one observes that, for each day (different symbols) the values of ${\mathrm{avg}}_{j}\left\{\langle \log_{2}\left[\sigma_{j}\right]\rangle \right\}$ at the different locations (different colours) are very similar. Indeed, the amplitude of the time-series is mainly due to the variation in the common signal around 50 Hz. Interestingly, the value of $H_{\infty }$ for each location seems to stay the same for different time intervals, i.e., for KA, OL, PT, and TU, respectively, at close to 2.4 bits, 2.8 bits, 3.1 bits, and 3.7 bits. This ordering of $H_{\infty }$ can be understood from the locations in the grid where the recordings were performed. It was found that the frequency fluctuations at one bus from the common overall trend are determined by the inverse of its resistance centrality in the grid [72]. One therefore expects recordings at the periphery of the grid to be more impacted by noise than those closer to the centre of the grid. Both KA and OL are rather central in the grid, which suggests that these recordings are essentially in line with the common trend around 50 Hz. On the other hand, PT and TU are peripheral in the grid, which suggests that their recordings are more impacted by randomness around the common trend. Thus, the Rényi min-entropy should be smaller for KA and OL compared to PT and TU, which is what can be observed in Figure 8.

## 4. Discussion

For many real-world systems, the intrinsic dynamics are so complex that inferring a mathematical model for the underlying dynamical process generating signal measurements becomes a very challenging task—if feasible at all. An alternative approach is to analyse the evolution of the information contained in the signals one can measure. While this approach fails to provide a detailed model for the microscopic dynamics, if successful, then it allows for different dynamical regimes and changes in the parameters of the system to be characterised.

With this in mind, here we propose to include the magnitude variability in the signal in the ordinal pattern (OP) encoding. Our aim is to complement the permutation entropy analysis, to improve the characterisation of the dynamical behaviours observed from the time-series, and to maintain explainable results. For example, if instead of using our complementary variables, we combine them, then there would be cases when the resultant modified PE value could not tell us whether it is increased (or decreased) because the PE value increased or because the standard deviations of the amplitudes increased (or decreased).

We first tested our approach on synthetically generated signals from chaotic bidimensional mappings under different parameter values. Specifically, we used coupled logistic maps (Equation (Equation 1)) and the Hénon map (Equation (Equation 2)). We analysed the signals coming from one of the two coordinates available for these maps because the OP encoding can only be applied to univariate signals.

The main reason to choose bidimensional mappings is that there are proofs showing that the permutation entropy of one-dimensional mappings directly relates to the Kolmogorov–Sinai entropy [73,74], which is not the case for higher-dimensional mappings. Because of this difference, it is expected that the characterisation of a dynamical regime by the permutation entropy for a higher-dimensional system is incomplete. We show that this problem is mitigated by including the standard deviation of the signal (in the embedded vectors forming the OPs) to the permutation entropy quantification (Figure 2 and Figure 4). We also find that our approach improves the characterisation of the dynamical regimes even under significant levels of observational noise and different choices of embedding dimension (Figure 5 and Figure 6).

We then analysed real-world EEG signals registered intracranially from 11 rats under free conditions throughout the sleep–wake cycle. We considered these signals because they come from a system—the brain—where the underlying microscopic dynamics are unknown (i.e., we lack a differential equation that models the system), there are inherent noise sources affecting the quality of the signal measurements, and the system has been hypothesised to have some level of chaoticity [75]. Moreover, in practice, the polysomnographic classification of sleep-wake states requires highly trained professionals to recognise characteristic electrophysiological patterns that vary according to the sleep stage, plus depend on anatomy and individual variability. However, the electrophysiological variability introduced by experimental manipulations (in research settings) or disease (in clinical settings), requires that whichever automatic sleep scoring classifier used is interpretable and not a black box.

Our results show that, for most cortical locations, independently of the embedding dimension used and the natural inter-animal variability, the states of active wakefulness, rapid-eye movement (REM) sleep, and non-REM sleep can be distinguished in the plane formed by the Rényi min-entropy and signal variability (Figure 7).

Finally, we analysed frequency recordings in the European electric power grid. We found that the standard deviation of the signals at different locations is similar during the same time intervals. This is expected, as the frequency in the whole grid essentially fluctuates around a common value close to 50 Hz with similar fluctuation magnitudes. We also found that the permutation entropy is higher for recordings at the periphery of the grid compared to those closer to the centre, which can be explained by the centrality of the recorded buses. Overall, our method identified the important features of the grid frequency dynamics.

Our analyses are limited in terms of the choice of embedding delay and overlap between consecutive embedded vectors; namely, for all our results τ=1 and the embedded vectors are quasi-non-overlapping, only sharing a single data point from the signal. The reason to restrict τ to 1 is that we can consider the entire signal, whereas τ>1 implies consideration of sub-sampled signals. On the other hand, increasing the overlapping of the embedded vectors creates artificial correlations between consecutive OPs (which are irrelevant for most permutation entropy calculations, but can affect conditional or transfer entropy calculations). For example, with a larger overlap than the one we choose here, two consecutive embedded vectors with D=3 would share two points, such that $\left\{x_{t},x_{t+1},x_{t+2}\right\}$ and $\left\{x_{t+1},x_{t+2},x_{t+3}\right\}$. This would imply that, if $x_{t+1}>x_{t+2}$, then the OP for the second embedded vector would be conditioned to this increasing relation that is present in the previous embedded vector, and all other OP possibilities would be forbidden (i.e., the OP would be artificially forced to take on particular OP symbols). Consequently, our choice of encoding parameters allows us to keep all the signal and have a maximum number of embedded vectors with null redundancy between consecutive ones.

Finally, we note that a natural extension to our approach could consider a three-dimensional space, composed of the permutation entropy, the signal variability within the embedded vectors, and a complexity measure, such as the Jensen–Shannon complexity, which could improve the characterisation of other dynamical regimes (such as non-chaotic ones). Moreover, in practical applications where the signals are short, instead of considering the standard deviation of the signal for the embedded vectors, one could consider the inter-quartile range, which is a statistically robust descriptor that is unaffected by outliers.

## Figures and Tables

**Figure 1 entropy-27-00840-f001:**
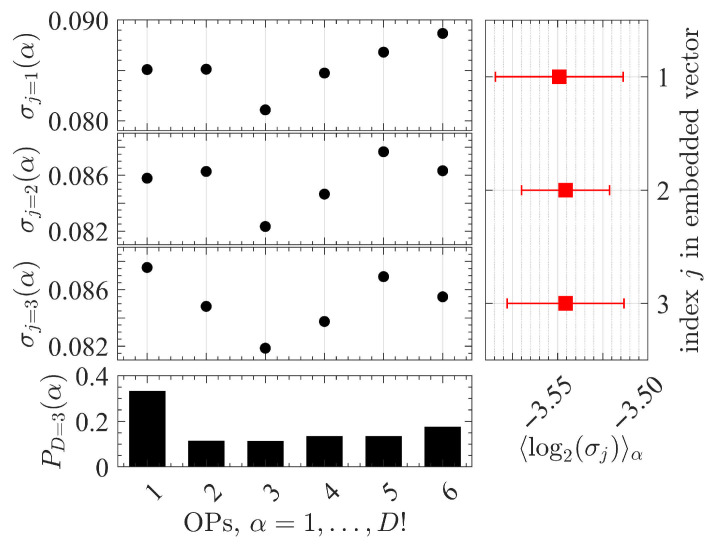
**Example of the variability of EEG signal magnitudes within each ordinal pattern (OP)**. The EEG signal is from the right primary motor cortex (rM1) of a representative rat during active wakefulness and the OPs are constructed using an embedding dimension D=3 and delay τ=1. The top three left panels show the values of the standard deviation of the EEG signal $\sigma_{j}\left(\alpha \right)$ in the components (j=1,2,3) of the embedded vectors for each OP symbol (α=1,…,6). The bottom left panel shows the OP probability distribution ${\left\{P\left(\alpha \right)\right\}}_{\alpha =1}^{D!}=\left\{P\left(1\right),\ldots ,P\left(D!=6\right)\right\}$. The right panel shows the mean (red squares) with respect to ${\left\{P\left(\alpha \right)\right\}}_{\alpha =1}^{6}$ for each set of $\log_{2}\left[\sigma_{j}\left(\alpha \right)\right]$ values, $\langle \log_{2}\left[\sigma_{j}\right]\rangle$, with error bars defined by $\pm \sqrt{\langle \log_{2}{\left[\sigma_{j}\right]}^{2}\rangle -{\langle \log_{2}\left[\sigma_{j}\right]\rangle }^{2}}$.

**Figure 2 entropy-27-00840-f002:**
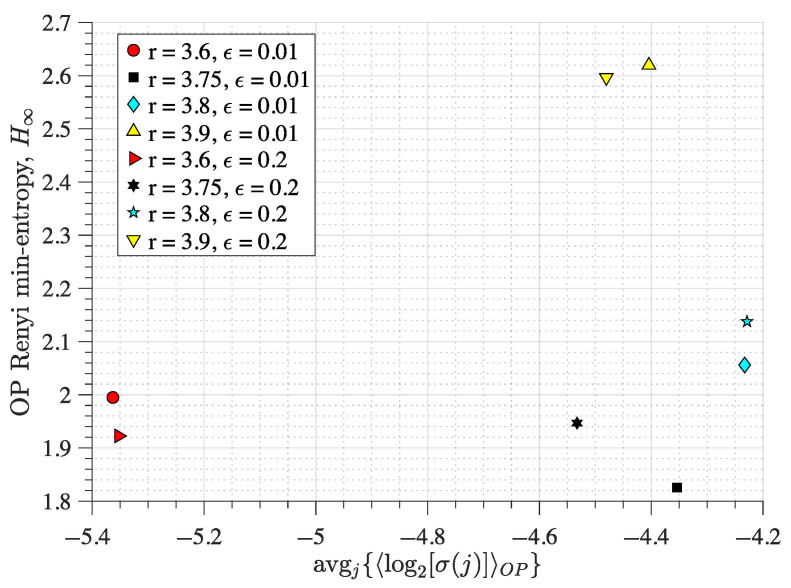
**Rényi min-entropy and average magnitude variability of the ordinal pattern (OP) embedded vectors from two coupled identical logistic maps**. The map iterates for the OP encoding are obtained from Equation (Equation 1), where the coupling strength ε is set to 0.01 or 0.2 and the map parameter *r* is set to 3.6 (red symbols), 3.75 (black symbols), 3.8 (cyan symbols), or 3.9 (yellow symbols). We use D=4 and τ=1 for the OP encoding of the iterates of the *x* component (one map)—see Section 2 for details.

**Figure 3 entropy-27-00840-f003:**
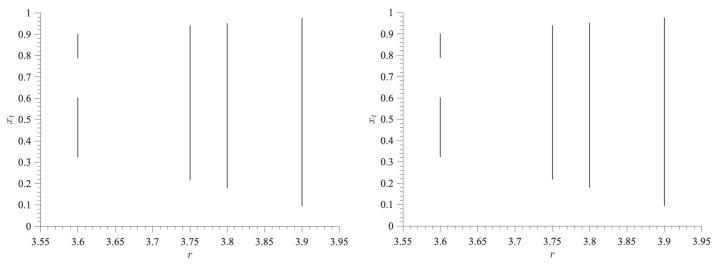
**Bifurcation-like diagrams for coupled identical logistic maps**. The left [right] panel shows the signal of one map, $x_{t}$ (Equation (Equation 1)), when ε=0.01 [ε=0.2] as *r* is changed according to the values used in Figure 2.

**Figure 4 entropy-27-00840-f004:**
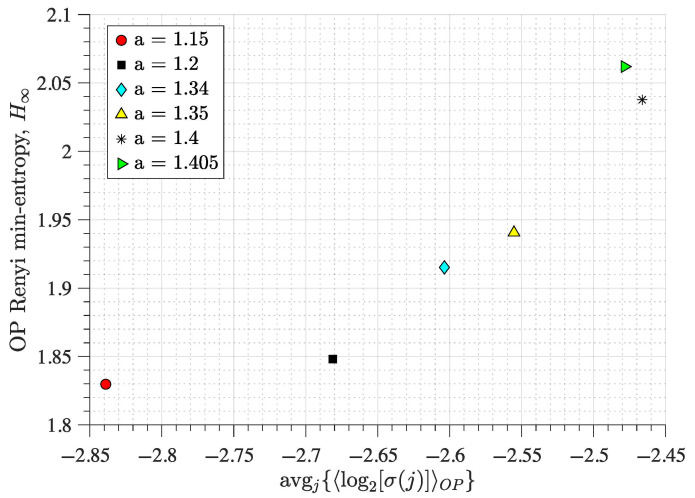
**Rényi min-entropy and average magnitude variability of the ordinal pattern (OP) embedded vectors from a Hénon map**. The map iterates are obtained from Equation (Equation 2) with b=0.3 and the other parameter set to either a=1.15 (red circle), 1.20 (black square), 1.34 (cyan diamond), 1.35 (yellow triangle), 1.40 (black asterisk), or 1.405 (green triangle). We use D=4 and τ=1 for the OP encoding of the iterates of the *x* component (as in Figure 2)—see Section 2 for details.

**Figure 5 entropy-27-00840-f005:**
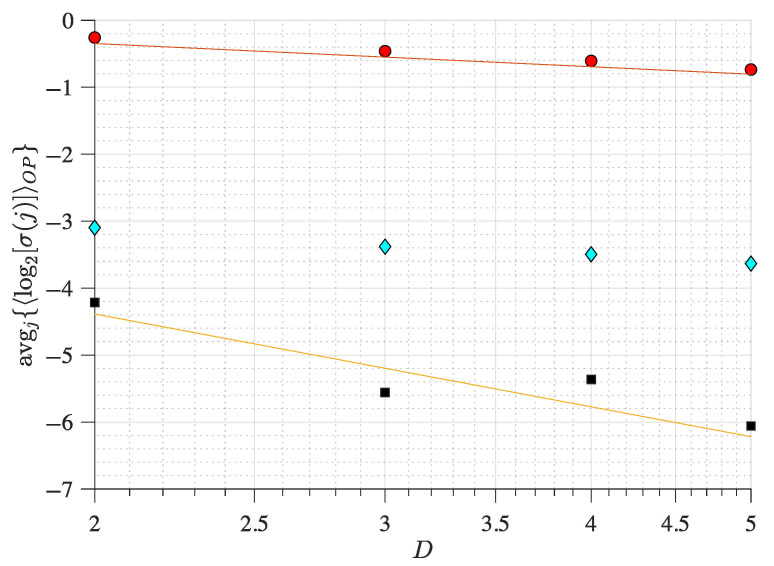
**Average magnitude variability of the ordinal pattern sequences from two coupled identical logistic maps as a function of the embedding dimension *D* and noise strength**. Black squares correspond to noiseless iterates, cyan diamonds to observational noise with a standard deviation of $10^{-1}$, and red circles to observational noise with a standard deviation of $10^{0}$. The map parameters for all symbols are set such that r=3.6 and ε=0.01 (Equation (Equation 1)). Two reference lines are included with slopes of −2 (orange) and −1/2 (red).

**Figure 6 entropy-27-00840-f006:**
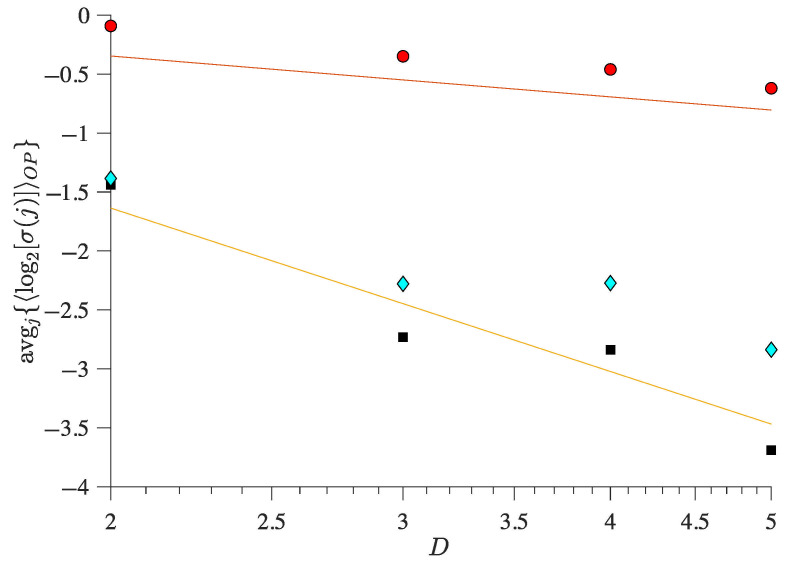
**Average magnitude variability of the ordinal pattern (OP) sequences from a Hénon map as a function of the OP embedding dimension and noise strength**. Filled symbols have the same observational noise as in Figure 5. The map parameters of the Hénon map are b=0.3 and a=1.15 (Equation (Equation 2)). Two reference lines are included with slopes of −2 (orange) and −1/2 (red).

**Figure 7 entropy-27-00840-f007:**
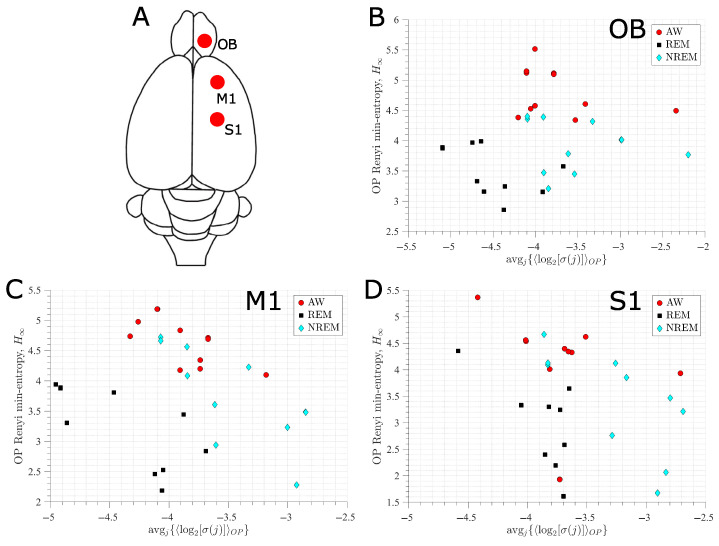
OPs analysis for EEG recordings from 11 rats in three sleep–wake states: active wakefulness (AW), rapid eye movement (REM) and non-REM (NREM) sleep. (**A**) Location of the electrodes for the EEG recordings corresponding to the right olfactory bulb (OB), and the primary motor (M1) and somatosensory (S1) cortices. (**B**–**D**) Rényi entropy vs. average magnitude variability, respectively, for BO, M1, and S1 electrodes. The embedding dimension for the OPs is D=5.

**Figure 8 entropy-27-00840-f008:**
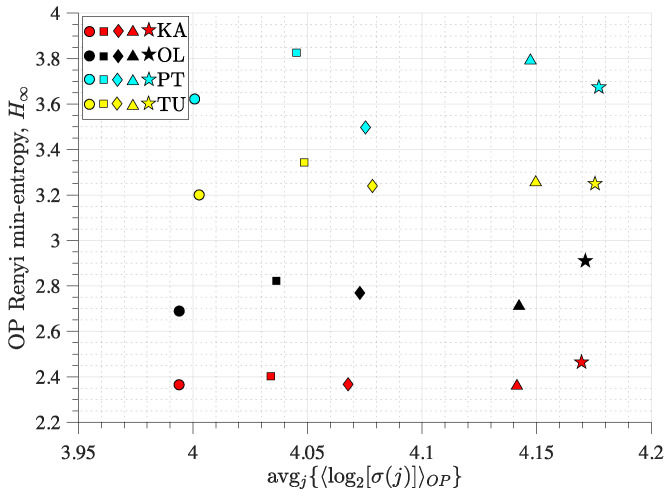
OPs analysis for GPS-synchronised grid frequency recording from 4 different locations in the European synchronous electric power grid: Karlsruhe (KA), Oldenburg (OL), Lisbon (PT), Istanbul (TU). Rényi min-entropy vs. average magnitude variability for the different locations and five different 24 h periods of recordings (respectively corresponding to the five different symbols). The embedding dimension for the OPs is D=5.

**Table 1 entropy-27-00840-t001:** Maximum Lyapunov exponent λ (MLE) of the coupled logistic maps for the parameters in Figure 2. These are found by using Wolf’s et al. method [68], using an embedding dimension of 2, a time delay of 1, a maximum number of iterations of 10 to track for divergence (which avoided saturation), and a 10% minimum separation to avoid temporal neighbours.

(r,ε)	(3.6,0.01)	(3.75,0.01)	(3.8,0.01)	(3.9,0.01)
λ	0.16	0.39	0.42	0.45
(r,ε)	(3.6,0.2)	(3.75,0.2)	(3.8,0.2)	(3.9,0.2)
λ	0.18	0.36	0.43	0.49

**Table 2 entropy-27-00840-t002:** Maximum Lyapunov exponent λ (MLE) of the Hénon map for the parameters in Figure 4. These are found as shown in Table 1.

*a*	1.15	1.2	1.34	1.35	1.4	1.405
λ	0.27	0.30	0.36	0.37	0.42	0.41

## Data Availability

No new data were created or analyzed in this study. Data sharing is not applicable to this article.

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
