# Peer review of "Including the Magnitude Variability of a Signal in the Ordinal Pattern Analysis"

_entropy, 2025, doi:10.3390/e27080840_

Round 1

Reviewer 1 Report

Comments and Suggestions for Authors
  • In the Introduction section, the authors did not explain the rationale for introducing standard deviation (std) information into permutation entropy (PE). What is the motivation behind this design?
  • In recent years, numerous improved PE algorithms have been proposed, such as WPE(Weighted-permutation entropy: A complexity measure for time series incorporating amplitude information, PHYSICAL REVIEW E87, 022911 (2013)), mPE (Modified permutation-entropy analysis of heartbeat dynamics, PHYSICAL REVIEW E 85, 021906 (2012)), AAPE (Hamed,Azami,Javier,et al.Amplitude-aware permutation entropy: Illustration in spike detection and signal segmentation.[J].Computer methods and programs in biomedicine, 2016.), and IPE (Improved Permutation Entropy for Measuring Complexity of Time Series under Noisy Condition, Complexity,Volume 2019, Article ID 1403829). How does the proposed method differ from these existing approaches? The authors are advised to elaborate on its unique advantages in the revised manuscript.
  • Line 142: The expression T/D should be corrected to T/D!
  • Figures 1 and 4: The references "– see Sect. ?? for details" are incomplete. Please provide the correct section numbers or descriptions.
  • The paper only discusses noiseless map iterates. However, real-world data often contain noise. It is recommended that the authors analyze a noisy model to test the algorithm’s robustness.
  • The validation relies solely on EEG signals, which may not sufficiently demonstrate the method’s general applicability. Additional real-world datasets, such as heart rate signals, fault diagnosis signals, or others, should be included to strengthen the conclusions.
  • A critical limitation of this work is the absence of benchmarking against existing advanced methods. To strengthen the study’s impact, the authors should rigorously compare their results with recent state-of-the-art techniques in the field

Author Response

Please see the attachment (reviewer 1)

Reviewer 2 Report

Comments and Suggestions for Authors

Report on the manuscript ID: Entropy-3687327

I have reviewed the manuscript titled “Including the magnitude variability of a signal into the ordinal pattern analysis”, authored by M. Tyloo, J. González, and N. Rubido. The authors propose to include the amplitude information of a time series under study in an ordinal characterization of its dynamical properties. To do so, they introduce a new metric that weights the logarithm of the standard deviation of the data at the j-component of all ordinal patterns. Applications to two chaotic maps and EEG records are included to show the strength as a classification metric for different dynamical behaviors.

The manuscript is well written, structured, and relevant to data-driven approaches and deserves to be published in Entropy, but not in the present form. Before its acceptance, some points should be clarified:

1.- I was surprised that the introduction lacks a review of the efforts that have been made to include the amplitude information of the time series into de ordinal patterns encoding. I strongly believe that a small paragraph/sentence must be included in the introduction to acknowledge previous works such as:

  • Fadlallah et al. Weighted-permutation entropy: A complexity measure for time series incorporating amplitude information. Phys. Rev. E. (2013) 87, 022911.
  • Azami et al. Amplitude-aware permutation entropy: Illustration in spike detection and signal segmentation. Comput. Methods Programs Biomed. (2016) 28, 40–51.
  • Zanin, M. Continuous ordinal patterns: Creating a bridge between ordinal analysis and deep learning. Chaos (2023) 33, 033114.
  • Olivares et al. Quantifying deviations from Gaussinity with applications fo flight delay distributions. Entropy (2025) 27(4), 354.

2.- The Eq. (4) shows the definition of the Rényi Min-entropy. The notation reads H_{\infty}. It must be clarified that the subscript $\infty$ means that the parameter $q$ (that defines the Renyi entropy) tends to infinity.

3.- The results depicted in Fig. 2 and 3 clearly show some dynamical differences for the different values of the parameter r. Yet, a more detailed discussion should be included. For instance, why the authors do not compare with the Lyapunov exponent? Are the differences due to different degrees of chaoticity?

4.- The use of bi-dimensional chaotic maps is justified because the permutation entropy of one-dimensional maps directly relates to the Kolmogorov-Sinai entropy. Nevertheless, the authors use only one coordinate of the equations. Why? Shouldn't they have used both coordinates and compared them?

5.- In line 207. The authors could include a reference where the bifurcation diagram of the Henon Map can be seen.

6.- From the results shown in Figs. 5 and 6. Is it possible to build a procedure to quantify the amount of noise contamination?

7.- In lines 231 and 283. I believe that “permutation entropy” should be replaced by Renyi min-entropy.

8.- In line 273, the sentence “… and it has been hypothesized to have some level of chaoticity” needs a reference.

9.- In the last paragraph of page 10, the authors have justified using the lag and the embedding dimension. The value of $\tau=1$ is justified when analyzing maps. However, for the EEG measurements, it must be understood that you are characterizing high-frequency correlations/temporal information when $\tau=1$. On the other hand, The forbidden OP possibilities that the authors mentioned are only valid when studying ordinal transitions and not when counting the occurrence of ordinal patterns.

Author Response

Please see the attachment (reviewer 2)

Round 2

Reviewer 1 Report

Comments and Suggestions for Authors

The author has addressed all my concerns, and I recommend accepting it for publication in its present form.

Reviewer 2 Report

Comments and Suggestions for Authors

All right